# Influence of Pitch on Surface Dose Distribution and Image Noise of Computed Tomography Scans

**DOI:** 10.3390/s23073472

**Published:** 2023-03-26

**Authors:** Kenji Yamada, Yoshiki Kawata, Masafumi Amano, Hidenobu Suzuki, Masahide Tominaga, Motoharu Sasaki, Hikaru Nishiyama, Masafumi Harada, Noboru Niki

**Affiliations:** 1Division of Clinical Technology, Tokushima University Hospital, Tokushima 7708503, Japan; 2Institute of Post-LED Photonics, Tokushima University, Tokushima 7708506, Japan; 3Department of Diagnostic Radiology, Graduate School of Biomedical Sciences, Tokushima University, Tokushima 7708503, Japan; 4Department of Therapeutic Radiology, Graduate School of Biomedical Sciences, Tokushima University, Tokushima 7708503, Japan; 5Department of Radiological Technology, Ehime University Hospital, Toon 7910295, Japan; 6Department of Radiology and Radiation Oncology, Tokushima University, Tokushima 7708503, Japan; 7Faculty of Science and Technology, Tokushima University, Tokushima 7708506, Japan

**Keywords:** helical computed tomography, pitch, surface dose distribution, FBP, IMR, image noise, power spectrum

## Abstract

This study evaluated the effect of pitch on 256-slice helical computed tomography (CT) scans. Cylindrical water phantoms (CWP) were measured using axial and helical scans with various pitch values. The surface dose distributions of CWP were measured, and reconstructed images were obtained using filtered back-projection (FBP) and iterative model reconstruction (IMR). The image noise in each reconstructed image was decomposed into a baseline component and another component that varied along the z-axis. The baseline component of the image noise was highest at the center of the reconstructed image and decreased toward the edges. The normalized 2D power spectra for each pitch were almost identically distributed. Furthermore, the ratios of the 2D power spectra for IMR and FBP at different pitch values were obtained. The magnitudes of the components varying along the z-axis were smallest at the center of the reconstructed image and increased toward the edge. The ratios of the 3D power spectra on the f_x_ axis for IMR and FBP at different pitch values were obtained. The results showed that the effect of the pitch was related to the component that varied along the z-axis. Furthermore, the pitch had a smaller effect on IMR than on FBP.

## 1. Introduction

Computed tomography (CT) is used to detect small changes in biological tissues in the early diagnosis of medical conditions. Although the popularity of CT is increasing, there is a risk of increased radiation exposure [1,2]. Helical scanning, which uses a helical trajectory, is a common high-speed scanning method. Images obtained through helical scanning in the body direction (z-axis) and cross-section (x–y plane) do not provide an even scan of an object positioned within the gantry [3,4,5,6,7,8,9,10,11,12,13,14,15,16,17]. It is well known that the image quality of a reconstructed image varies with the pitch of the helical scan [4,5]. Numerous studies have been conducted to analyze and improve reconstructed images obtained from helical scans. Hara et al. [7] highlighted image quality inhomogeneity using the modulation transfer function (MTF), noise power spectrum (NPS), and signal-to-noise ratio (SNR) analysis of the center and edges of the axial section of three multidetector computed tomography (MDCT) scans. Frederic et al. [8] measured and evaluated the effects of reconstruction algorithms and filters on 64-slice and 128-slice helical CT scans using local 2D and 3D noise spectra. Li et al. [9] experimentally evaluated the unique noise characteristics of a statistical model-based iterative reconstruction (MBIR) method using a clinical CT system. The noise characteristics were found to be significantly different from those obtained using filtered back-projection (FBP). Greffier et al. [10] compared the noise magnitude and texture computed from the NPS across two generations of iterative reconstruction (IR) algorithms proposed by three manufacturers based on the dose level. Lambert et al. [11] evaluated whether the axial mode or helical mode is more appropriate for a 16 cm collimation CT scanner that is capable of step-and-shoot volumetric axial coverage. The results showed that helical scanning was more effective for scan lengths above 16 cm. Szczykutowicz et al. [16] analyzed changes in the CT number and image noise as functions of patient position within a 64-slice CT scanner. Absolute CT numbers of over 20 HU were observed at phantom positions 10 cm from the isocenter. Noise uniformity varied by over two-fold 10 cm below the isocenter. 

This study analyzed the effect of pitch on helical scans, using surface dose distribution, FBP image noise, iterative model reconstruction (IMR) image noise, and their power spectra for three clinical pitches using a 256-slice CT scanner and a cylindrical water phantom (CWP). The results show that the surface dose distribution and image noise in the z-axis direction exhibit a pitch-dependent phase difference. Furthermore, the image noise was analyzed by being decomposed into a baseline component and another component that varied along the z-axis. The effect of the pitch was related to the component that varies along the z-axis. Compared to that of FBP, the pitch had a smaller effect on IMR. In addition, the effect of pitch on helical CT was clarified.

## 2. Materials and Methods

### 2.1. Overview of Measurement and Analysis

Figure 1 shows an overview of the measurement and analysis procedures. 

An ethical review is not required for this clinical trial, as it does not involve medical research on human or animal subjects. A CWP was used to measure the surface dose distributions and projection data based on one axial scan and three helical scans with pitch values of 0.996, 0.696, and 0.399. FBP and IMR were used for image reconstruction. The relationships between the surface dose distribution, image noise, and power spectrum were quantitatively evaluated for each pitch. Furthermore, the image noise was analyzed through decomposition into a baseline component and another component that varied along the z-axis.

### 2.2. Measurement

The CWP (Canon Medical Systems Inc., Tokyo, Japan) was 320 mm in diameter and 250 mm in length. A 256-slice CT scanner (Brilliance iCT, Philips Healthcare, Amsterdam, The Netherlands) was used to perform axial and helical scans. The scan parameters included a tube voltage of 120 kV, an effective tube current of 370 mA s/slice, a rotation speed of 0.5 s/rotation, a FOV diameter of 400 mm, a collimation width of 0.625 mm, 128 detector rows, beam width of 80 mm (0.625 mm × 128 rows), and pitch values of 0.992, 0.696, and 0.399. The reconstruction algorithms were based on the reconstruction kernel standard B for FBP [4] and the image definition routine with level-1 noise reduction for IMR [18], which are clinical scanning protocols used for soft tissue regions. The reconstructed images were 1.0 mm thick with a reconstruction interval of 1.0 mm. The film dosimeter [19,20,21,22,23,24] used to measure surface dose distribution in the z-axis direction comprised a Gafchromic film (XR-SP2, Ashland Ltd., Wilmington, NJ, USA) and a film scanner (Epson Expression 11000G flat-table document scanner and “DD-system,” R-Tech Inc., Tokyo, Japan). Surface dose distributions were analyzed using software (DD-Analysis Ver.10.33, R-Tech Inc., Tokyo, Japan) on a personal computer (Windows 7; R-Tech Inc., Tokyo, Japan). To simultaneously measure the surface dose distribution and reconstructed images, a Gafchromic film was attached to the CWP surface, and a metal sphere with a diameter of 0.5 mm was attached to the film for alignment. Figure 2 shows an overview of the CWP mounted on a 256-slice CT scanner table. 

Scanning experiments were performed as follows.
A fixture was used to secure the CWP to the table.The central CWP axis was aligned with the central CT scanner gantry axis using a 256-slice CT positioning laser. The film was attached at the center positions (0, 160, 0) to (0, 160, 250) on the surface of the CWP, and metal spheres were attached at positions (0, 160, 10) and (0, 160, 240) on the film to mark the start and end scan positions, respectively.The projection data for an axial scan and three helical scans with pitch values of 0.996, 0.696, and 0.399 were reconstructed using FBP and IMR.

To evaluate the measurement accuracy of the Gafchromic film dosimeters, the average dose per unit area of the film dosimeter and the average dose per unit volume of the ionization chamber dosimeter (Accu-Gold and 10 X 6–0.6 CT multislice chamber, RADCAL CORPORATION, Monrovia, CA, USA) were measured 10 times at each pitch. The average dose in the case of the film dosimeters was measured as the average dose for a 9 × 100 mm^2^ area of the Gafchromic film on the CWP surface. Tominaga et al. [22] converted the film density of a Gafchromic film to a dose. The film exhibited a local inhomogeneity of 15% and measurement uncertainty of 1.5% [25,26]. The ionization box dosimeter was placed at the same location on the CWP surface for measuring the average dose per unit volume.

### 2.3. Analysis

Measurements were performed 10 times each for the axial scan and the three helical scans with pitch values of 0.992, 0.696, and 0.399. The surface dose distributions were measured at intervals of 0.169 mm over a length of 180 mm along the z-axis. The density of the Gafchromic film was converted into a dose. The average value of ten measurements of the surface dose profile was used as the dose (z). The standard deviation of the pixels within 25 mm of the x–y plane for the 10 measurements centered on the pixel of interest (x, y, z) in the reconstructed image was calculated as the image noise (x, y, z). The image noise (x, y, z) is denoted as noise (x, y, z). Noise (x, y, z) was decomposed into a baseline component and another component that varies in the z-axis direction, denoted as Noise BL (x, y) and the Noise VA (x, y, z), respectively. Noise BL (*x*, *y*) is defined as the average noise (x, y, z) on the z-axis. Noise VA (x, y, z) is the value obtained by subtracting Noise BL (x, y) from noise (x, y, z). The power spectra of Noise BL (x, y) and Noise VA (x, y, z) were calculated as [27,28].
(1)$$\mathrm{PS}\;\mathrm{BL}\left(f_{x},\;f_{y}\right)=\frac{b_{x}b_{y}}{L_{x}L_{y}}{\left|\mathrm{FFT}\left[\mathrm{Noise}\;\mathrm{BL}\left(x,y\right)\right]\right|}^{2},$$
(2)$$\mathrm{PS}\;\mathrm{VA}\left(f_{x},\;f_{y},\;f_{z}\right)=\frac{b_{x}b_{y}b_{z}}{L_{x}L_{y}L_{z}}{\left|\mathrm{FFT}\left[\mathrm{Noise}\;\mathrm{VA}\left(x,y,z\right)\right]\right|}^{2},$$
where $b_{x},\;b_{y},\;b_{z}$ are the voxel sizes in the x, y, and z directions, respectively. $L_{x},\;L_{y},\;L_{z}$ are the lengths of each realization in the x, y, and z directions, respectively. Zero padding was employed to increase the length of the input image in the x, y, and z directions to a number that could be expressed as a power of 2. The Python library of NumPy for fast Fourier transform (FFT) calculations was used.

## 3. Result

### 3.1. Image Analysis

Table 1 summarizes the average doses for the film dosimeters and the average doses per unit volume for the ionization chamber dosimeters at pitch values of 0.992, 0.696, and 0.399. 

The ratios of the average doses of the film dosimeters to the average doses of the ionization chamber dosimeters with pitch values of 0.992, 0.696, and 0.399 were 1.013, 0.981, and 0.983, respectively. The accuracy of the film dosimeters was comparable to that of the ionization chamber dosimeters. Figure 3 and Figure 4 show the dose (z), noise (0, 135, z), noise (0, 90, z), noise (0, 45, z), and noise (0, 0, z) for the axial and helical scans reconstructed using FBP and IMR, respectively. 

The analysis regions of noise (0, 135, z), noise (0, 90, z), noise (0, 45, z), and noise (0, 0, z) are denoted as ROI1, ROI2, ROI3, and ROI4, respectively. Four measurement positions were selected in the radial direction of the rotationally symmetric cylindrical phantom. The axial scans in Figure 3a and Figure 4a show that noise (x, y, z) is larger at the edge of the scan range of approximately 85–165 mm (80 mm detector width). Furthermore, the three helical scans in Figure 3b–d and Figure 4b–d show that noise (x, y, z) varies periodically. At the same pitch, noise (x, y, z) exhibited the same phase, and the phases of dose (z) and noise (x, y, z) were pitch-dependent. High-dose regions of dose (z) corresponded to small noise (x, y, z) and large noise (x, y, z) regions at pitch values of 0.992 and 0.696, respectively. Table 2 shows the average values and standard deviations of dose (z), noise (0, 135, z), noise (0, 90, z), and noise (0, 45, z) in the z-axis direction for FBP and IMR. The average values and standard deviations of the axial scan were calculated for lengths of ±40 mm from the center plane of rotation. The three helical scans with pitch values of 0.996, 0.696, and 0.323 were calculated for each pitch length. The average value of noise (x, y, z) for the axial and helical scans decreased toward the edge of the CWP, while the standard deviation of noise (x, y, z) for the helical scan increased. The average values of noise (0, 135, z), noise (0, 90, z), noise (0, 45, z), and noise (0, 0, z) were nearly equal at each pitch, and the standard deviation of noise (x, y, z) increased toward the edge of the CWP as the pitch increased. The ratios of the average values of noise (0, 135, z), noise (0, 90, z), noise (0, 45, z), and noise (0, 0, z) of the IMR to those of the FBP of the three helical scans were 0.396 ± 0.011, 0.392 ± 0.013, 0.390 ± 0.016, and 0.380 ± 0.018, respectively; these ratios were almost equal. The standard deviation for IMR was smaller than that for FBP. Moreover, it increased with an increase in the pitch and as the measurement position approached the edge of the CWP.

### 3.2. Frequency Analysis

Figure 5 and Figure 6 show the 2D power spectra PS BL (*f_x_*, *f_y_*) and 3D power spectra PS VA (*f_x_*, *f_y_*, *f_z_*) for the axial and helical scans, respectively. Figure 5a,b show the PS BL (*f_x_*, *f_y_*) on the f_x_ axis and its 2D normalized power spectra for FBP. The calculations were performed on a 512 × 512 image with zero padding to an input image of 276 × 276. The 2D normalized spectra of the axial and helical scans exhibited the same shape. The spectra were rotationally symmetric on the *f_x_*–*f_y_* plane and exhibited a low-pass frequency response. Figure 5c,d show the PS VA (*f_x_*, *f_y_*, *f_z_*) along the *f_z_* axis and its 3D normalized power spectra for the three helical scans. Calculations for pitches 0.399, 0.696, and 0.996 were performed on 512 × 512 × 512 images with zero padding for the input images of 276 × 276 × 30, 276 × 276 × 55, and 276 × 276 × 80, respectively. These peaks were observed at the fundamental and harmonic frequencies of the pitch. As the pitch increased, the fundamental and harmonic frequencies decreased, and their magnitudes increased. Figure 6a,b show the PS BL (*f_x_*, *f_y_*) on the *f_x_* axis and its 2D normalized power spectra for IMR. The 2D normalized power spectra of the axial and helical scans exhibited the same shape. Figure 6c,d show the PS VA (*f_x_*, *f_y_*, *f_z_*) on the *f_z_* axis and its normalized power spectra for the three helical scans. When comparing IMR and FBP, it was found that the normalized power spectra of the PS BL (*f_x_*, *f_y_*) on the *f_x_* axis exhibited the same shape. The PS BL (*f_x_*, *f_y_*) values for IMR at pitches of 0.339, 0.696, and 0.969 were 0.140, 0.149, and 0.152 smaller than those for FBP, respectively. These ratios were approximately the same. The ratios of the PS VA (*f_x_*, *f_y_*, *f_z_*) for IMR to those for FBP at pitch values of 0.399, 0.696, and 0.996 were 0.0152, 0.0513, and 0.124, respectively.

## 4. Discussion

The influence of pitch on helical scans used in clinical practice was demonstrated by measuring and analyzing surface dose distributions and image noise using CWPs. The surface dose distribution dose (z) of the axial scan, shown in Figure 3a and Figure 4a, increased sharply at the edge of the z-axis scanning range (85 mm and 165 mm), which indicated that the anode side was smaller than the cathode side owing to the heel effect. The surface dose distribution Dose (z) of the helical scans, shown in Figure 3b–d and Figure 4b–d, formed a profile that showed the overlap of Dose (z) of the axial scan according to the pitch. The period of Dose (z) at each pitch is a product of the detector width and pitch. The average value of each Dose (z) was equal, and its standard deviation increased as the pitch increased. At pitch 0.992, the high-dose region of dose (z) corresponded to the low-dose region of noise (z), and Dose (z) and noise (z) were in opposite phases. At 0.696, the high-dose region of dose (z) and low-dose region of noise (z) shifted, and the high-dose region was higher than the high-dose regions of 0.996 and 0.399. The average values of noise (0, 135, z), noise (0, 90, z), and noise (0, 45, z) for the reconstructed images at each pitch decreased toward the edge of the CWP and were nearly equal, regardless of the pitch. Furthermore, its standard deviation increased toward the edges of the CWP and became larger as the pitch increased. IMR showed the same trend as that of FBP; however, the average values and standard deviations were much smaller.

Image noise was decomposed into a baseline component and another component that varied along the z-axis. 2D normalized power spectra (*f_x_*, *f_y_*) of PS BL showed the same rotationally symmetric low-pass characteristics for the axial and helical scans, and FBP and IMR were nearly identical. The baseline component was not affected by the pitch. 3D power spectrum PS VA (*f_x_*, *f_y_*, *f_z_*) exhibited peaks at the fundamental and harmonic frequencies of pitch on the *f_z_* axis. The fundamental and harmonic frequency components increased as the pitch increased. On comparing the magnitude of PS VA (*f_x_*, *f_y_*, *f_z_*) on the *f_z_* axis at each pitch, it was found that IMR and FBP were more affected as the pitch increased, while IMR was less affected than FBP. 

The experimental constraints were one each of CWP, CT, and the scan parameters (tube voltage/effective tube current). The Appendix A shows the results of the measurements using the same CT with a CWP of diameter 200 mm and an effective tube current of 200 mA. The influence of pitch was also analyzed on a 160-row detector CT (Aquilion ONE/Vision Edition, Canon Medical Systems Inc., Tokyo, Japan) using the above CWP and almost identical imaging protocols. Similar results were obtained for the two CT scanners. Further evaluation of Rando phantom and patient data, and of different CT and clinical scan parameters (such as tube voltage and effective tube current), are needed to validate the results of this research. The analysis method can be used to analyze the influence of pitch on various helical scans of different CTs. For image noise, the standard deviation of each pixel was calculated using images taken 10 times, but there is a method to estimate image noise by subtracting two images [29,30]. Comparative evaluation with this method is also needed.

## Figures and Tables

**Figure 1 sensors-23-03472-f001:**
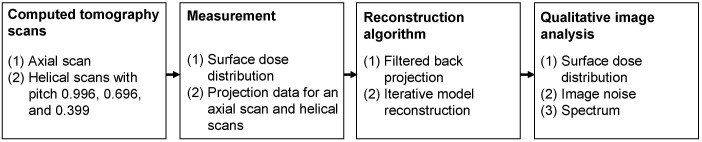
Measurement and analysis of the effect of pitch on 256-slice CT scans.

**Figure 2 sensors-23-03472-f002:**
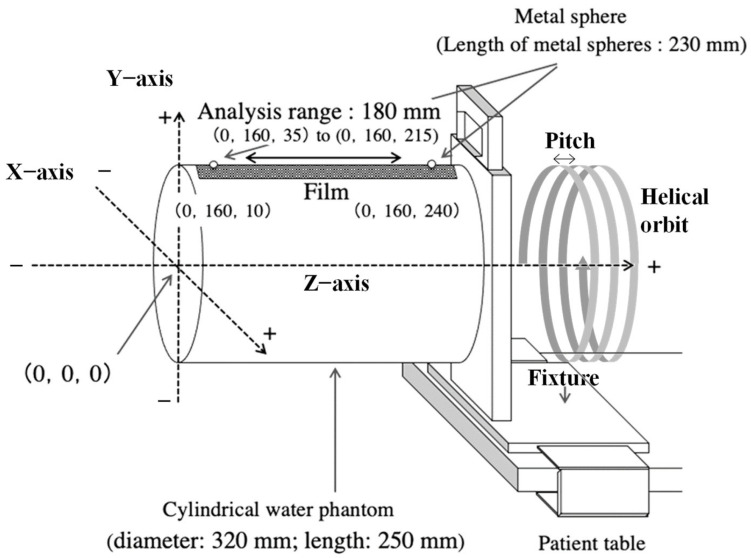
Overview of CWP on the multi-slice CT scanner table, with helical orbit and pitch.

**Figure 3 sensors-23-03472-f003:**
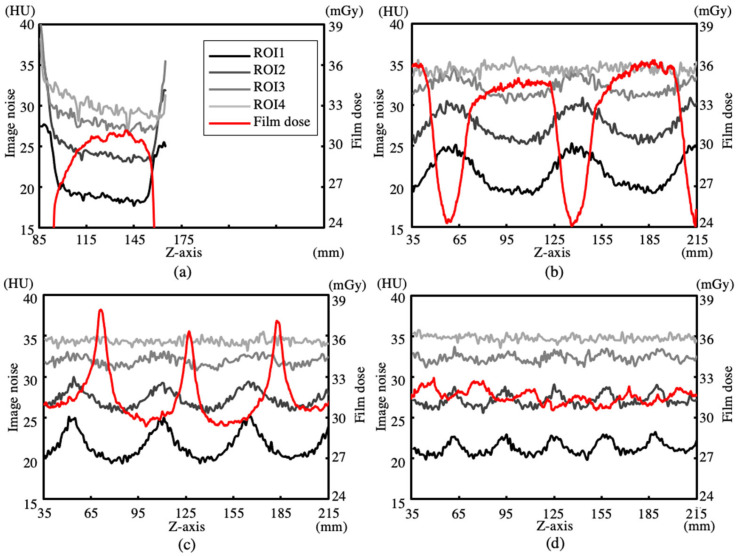
Dose (z) and Noise (x, y, z) for ROI1, ROI2, ROI3, and ROI4 in FBP images. (**a**) Axial scan. Helical scans with pitch values of (**b**) 0.992, (**c**) 0.696, and (**d**) 0.399.

**Figure 4 sensors-23-03472-f004:**
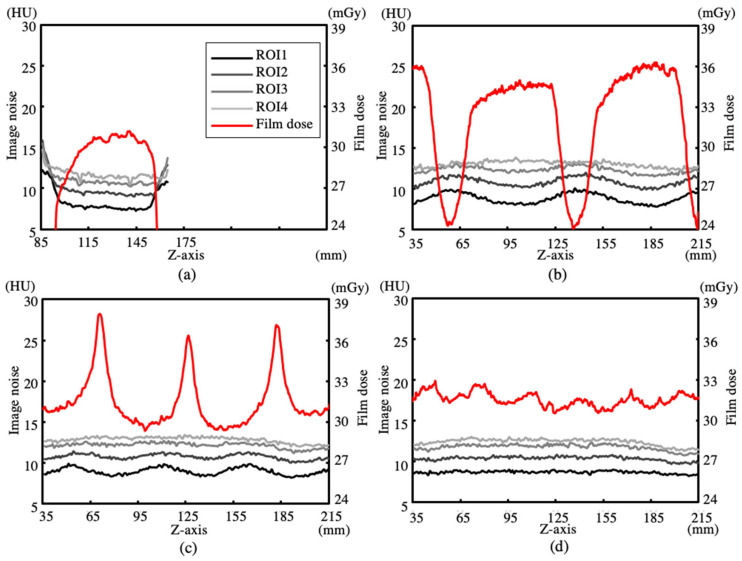
Dose (z) and Noise (x, y, z) for ROI1, ROI2, ROI3, and ROI4 in IMR images. (**a**) Axial scan. Helical scans with pitch values of (**b**) 0.992, (**c**) 0.696, and (**d**) 0.399.

**Figure 5 sensors-23-03472-f005:**
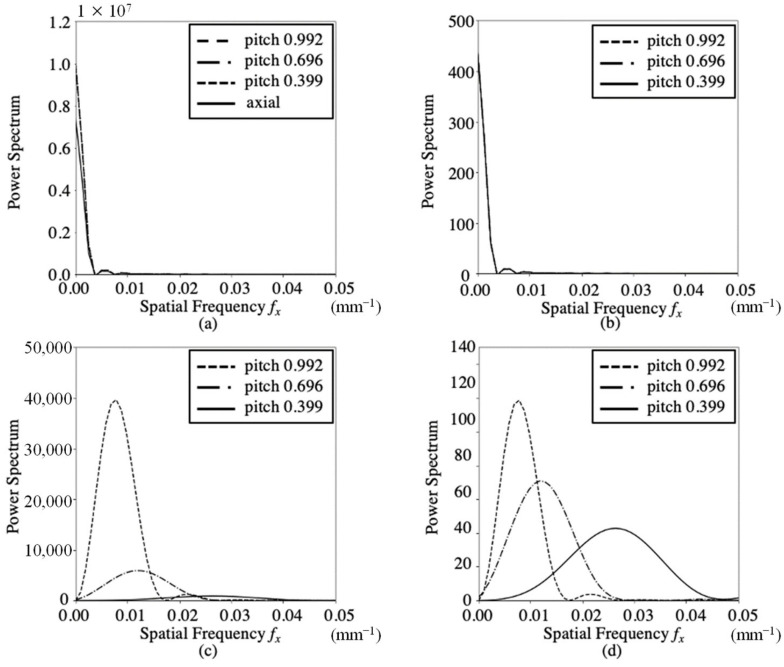
2D and 3D spectra of Noise BL (*x*, *y*) and Noise VA (*x*, *y*, *z*) in FBP images for axial and helical scans. (**a**) 2D spectra of Noise BL (*x*, *y*) on the *f_x_* axis, (**b**) 2D normalized spectra of Noise BL (*x*, *y*) on the *f_x_* axis, (**c**) 3D spectra of Noise VA (*x*, *y*, *z*) on the *f_x_* axis, and (**d**) 3D normalized spectra of Noise VA (*x*, *y*, *z*) on the *f_x_* axis.

**Figure 6 sensors-23-03472-f006:**
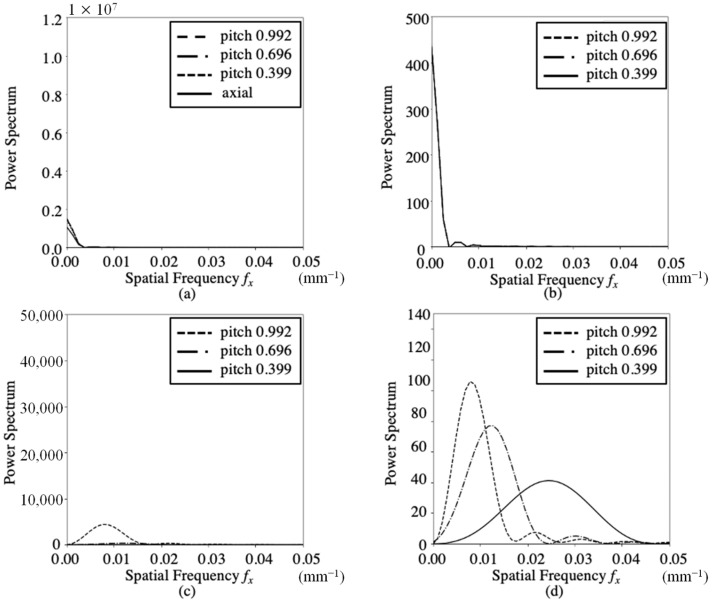
2D and 3D spectra of Noise BL (*x*, *y*) and Noise VA (*x*, *y*, *z*) in IMR images for axial and helical scans. (**a**) 2D spectra of Noise BL (*x*, *y*, *z*) on the *f_x_* axis, (**b**) 2D normalized spectra of Noise BL (*x*, *y*, *z*) on the *f_x_* axis, (**c**) 3D spectra of Noise VA (*x*, *y*, *z*) on the *f_x_* axis, and (**d**) 3D normalized spectra of Noise VA (*x*, *y*, *z*) on the *f_x_* axis.

**Table 1 sensors-23-03472-t001:** Comparison of the measurement performance of the film dosimeter and ionization chamber dosimeter with pitch values of 0.992, 0.696, and 0.399.

	Film Dosimeter		Ionization Chamber Dosimeter		Average Film Dose/Average Ionization Chamber Dose (%)
Pitch	Average Film Dose (mGy)	S.D.	Average Ionization Chamber Dose (mGy)	S.D.	
0.992	32.81	1.151	32.40	0.210	1.013
0.696	31.80	0.828	32.42	0.188	0.981
0.399	31.64	0.627	32.18	0.013	0.983

S.D.: standard deviation.

**Table 2 sensors-23-03472-t002:** Average values and standard deviations for ROI1, ROI2, ROI3, and ROI4 of reconstructed images by FBP and IMR for the axial and helical scans.

Pitch		Surface Dose (mGy)	Image Noise (HU)							
			FBP				IMR			
			ROI1	ROI2	ROI3	ROI4	ROI1	ROI2	ROI3	ROI4
Axial scan	Profile average	27.06	20.42	25.47	28.76	30.23	8.44	10.10	11.09	11.68
	Profile S.D.	5.53	2.94	3.23	2.48	1.70	1.37	1.38	0.95	0.62
Helical scan with a pitch of 0.992	Profile average	31.67	21.69	27.64	32.16	34.38	8.96	10.86	12.31	13.04
	Profile S.D.	4.11	1.97	1.62	0.98	0.48	0.63	0.53	0.40	0.37
Helical scan with a pitch of 0.696	Profile average	31.48	21.5	27.25	31.97	34.36	8.91	10.73	12.13	12.85
	Profile S.D.	2.00	1.59	1.04	0.56	0.37	0.45	0.34	0.32	0.32
Helical scan with a pitch of 0.399	Profile average	31.45	21.24	27.12	32.34	34.31	8.67	10.39	11.77	12.44
	Profile S.D.	0.50	0.80	0.73	0.51	0.35	0.17	0.25	0.38	0.39

FBP, filtered back-projection; HU, Hounsfield unit; IMR, iterative model reconstruction; SD, standard deviation; ROI, region of interest.

## Data Availability

All data generated or analyzed during this study are included in this published article.

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
