# Peer review of "Influence of Pitch on Surface Dose Distribution and Image Noise of Computed Tomography Scans"

_sensors, 2023, doi:10.3390/s23073472_

Round 1

Reviewer 1 Report

Comments:

In this manuscript, the authors evaluated the effect of pitch on helical scans of a 256-slice computed tomography (CT). The study is comprehensive. I would like to recommend its publication before below few questions are addressed.

1.       The font size of Fig. 1 is a little small.

2.       Is it possible to combine Fig. 3 and 4, Fig. 5 and 6, to make it more clear to show the comparison between IMR and FPB.

3.       Reference 28 has repeated numbering.

Reviewer 2 Report

This paper tried to understand the influence of pitch on the dose and noise of CT scanners. The paper is well-written and technically sound. Below are my questions and comments.

1. The authors picked 4 different profiles to measure. Could the authors justify why they picked these four profiles?

2. In figure 5 and figure 6, (c) and (d) are labeled as fz in the caption but fx in the figures. Could this be corrected?

Reviewer 3 Report

The authors present an interesting approach to evaluate the effect of pitch on helical scans of a 256-slice computed tomography (CT) scanner. The researchers used cylindrical water phantoms (CWP) and measured surface dose distributions of CWP and obtained reconstructed images using filtered back projection (FBP) and iterative model reconstruction (IMR) methods. The effect of pitch was related to the component that varied along the z-axis, and IMR had a smaller effect on the pitch than FBP. These findings could help optimize CT imaging protocols and improve image quality while reducing radiation dose to patients. This could be particularly relevant in cases where a high scan speed is required, such as in emergency or trauma situations, where the reduction in scan time could improve patient outcomes. Moreover, the study's findings could inform the selection of appropriate imaging parameters and reconstruction techniques to balance scan speed and image quality, thereby enhancing diagnostic accuracy and patient care.

There are some limitations to the above research. Some of the following issues need to be clarified before publish in this journal.

1. The study was conducted using a specific CT scanner model, and the results may not be generalizable to other scanner models or imaging protocols. Additionally, the study was conducted using cylindrical water phantoms, which may not fully represent the complexity and variability of human anatomy. Therefore, the studies using patient data are needed to validate the findings of this research.

2. There are many references that could be helpful for evaluating radiation dose in computed tomography like using a Rando phantom with a radiation dosimeter. Rando phantoms can be used to simulate a wide range of medical conditions and anatomical variations, allowing researchers to test imaging techniques in realistic scenarios. Why did you choose the cylindrical water phantoms (CWP)?

3. Automatic exposure control systems, such as automatic milliampere-seconds (auto-mA), are commonly used in CT to optimize radiation dose and image quality. However, there are some limitations to using auto-mA in radiation dose assessment. How to evaluate the accuracy and effectiveness of auto-mA systems in different patient populations or for different CT protocols?

4. CTDI is useful for comparing radiation dose between different scanners or protocols, how about your results compare with CTDI?

5. The study only evaluated the impact of pitch on image noise and did not assess other image quality parameters such as spatial resolution or contrast-to-noise ratio.

6. The study did not investigate the clinical impact of the observed differences in image quality between the two reconstruction methods. How to determine whether the observed differences translate into improved clinical outcomes for patients.

Reviewer 4 Report

Thank you for your submission. I found the paper generally easy to follow. Here are a few suggestions and questions:

-- I think defining "pitch" with a graphic or at least a general description would be helpful. Perhaps the pitch component could be added to Fig. 2.

-- Line 67: "...IMR had a smaller effect on the pitch." I think you mean "...the pitch had a smaller effect on the IMR". Isn't the pitch the thing you are varying to observe the effects?

-- Lines 133-134: "Noise BL is the average value of noise in the z-axis direction." Didn't you define the noise BL as being in the x,y directions?

-- Line 143: zero-padding increases the image size, it doesn't decrease it. Change "reduce" to "increase".

-- Fig. 5 (c) and Fig. 6 (a,c): The magnitude of the PSD is very large. Is the scaling correct? One has to be careful about "factors of N" that float around depending on what FFT function you are using. I'm not saying it's wrong, but one can double-check.

-- The conclusions of the paper didn't really jump out at me. A reader who is less familiar with CT scanning might want to know what the big takeaways are. Does this show some major drawbacks to using pitch scanning?

Round 2

Reviewer 3 Report

Authors have improved in this revised version. I have no other comments.